# A Study of Terahertz-Wave Cylindrical Super-Oscillatory Lens for Industrial Applications

**DOI:** 10.3390/s21206732

**Published:** 2021-10-11

**Authors:** Ayato Iba, Makoto Ikeda, Verdad C. Agulto, Valynn Katrine Mag-usara, Makoto Nakajima

**Affiliations:** 1Institute of Laser Engineering, Osaka University, Osaka 565-0871, Japan; iba.ac@om.asahi-kasei.co.jp (A.I.); vcagulto@ile.osaka-u.ac.jp (V.C.A.); valynn@ile.osaka-u.ac.jp (V.K.M.-u.); 2Asahi Kasei Corporation, Shizuoka 416-8501, Japan; ikeda.mm@om.asahi-kasei.co.jp

**Keywords:** terahertz, focusing, super-oscillatory lens

## Abstract

This paper describes the design and development of a cylindrical super-oscillatory lens (CSOL) for applications in the sub-terahertz frequency range, which are especially ideal for industrial inspection of films using terahertz (THz) and millimeter waves. Product inspections require high resolution (same as inspection with visible light), long working distance, and long depth of focus (DOF). However, these are difficult to achieve using conventional THz components due to diffraction limits. Here, we present a numerical approach in designing a 100 mm × 100 mm CSOL with optimum properties and performance for 0.1 THz (wavelength λ = 3 mm). Simulations show that, at a focal length of 70 mm (23.3λ), the focused beam by the optimized CSOL is a thin line with a width of 2.5 mm (0.84λ), which is 0.79 times the diffraction limit. The DOF of 10 mm (3.3λ) is longer than that of conventional lenses. The results also indicate that the generation of thin line-shaped focal beam is dominantly influenced by the outer part of the lens.

## 1. Introduction

The applications of terahertz (THz) and millimeter waves, including automotive radars and telecommunications, are expanding [1,2,3,4,5,6,7,8,9,10,11,12,13,14,15,16,17,18,19,20,21,22,23,24,25,26,27,28,29,30,31,32]. The THz frequency region is especially receiving attention in anticipation of the 6G technology. THz and millimeter waves are also expected to be highly effective for industrial product inspections. For instance, THz detection of submillimeter metal impurities embedded in a pressboard has been reported using THz time-domain spectroscopy [33]. Metal inclusions in opaque materials can usually be detected using X-rays; however, the use of ionizing radiation is not attractive for product inspection during the manufacturing process. Moreover, it will be highly expensive to upscale x-ray scanners for products with large areas, such as films, which take a longer time to measure. General inspection techniques with visible light utilize line cameras, cylindrical lenses, and one-dimensional scanning systems to reduce the inspection time by subjecting the product to a scanning motion. In contrast to visible light, THz waves are transparent to various materials, such as plastics and insulators, and are opaque to metals, which is practical for industrial inspection applications. However, the resolution is compromised due to the diffraction limit of THz waves, which have longer wavelengths than visible light. With THz waves, conventional lenses cannot achieve a resolution beyond the diffraction limit. Various methods have been studied to improve the resolution of measurements that employ THz waves. Many of them, such as the scanning near-field optical microscope and the laser THz emission microscope [3,34], take advantage of evanescent waves, severely limiting the working distance. In industrial inspection, it is often necessary to use long working distances (WD). Moreover, a long depth of field (DOF) is preferable so that the displacements perpendicular to the scanning direction can be ignored. Thus, one-dimensional measurements, which are useful in product inspection, require components with high resolution, long WD, and long DOF.

Focusing beyond the diffraction limit can be achieved using super-oscillatory lenses (SOL) [35,36]. SOL can focus into a hotspot in the far-field without the contributions from evanescent waves. It has also been reported to achieve a long DOF [37]. Though most SOLs are developed for visible light and UV, there are also some studies for the THz region [38,39,40]. Super-oscillatory lenses are defined lenses that use the physical phenomenon of “super oscillation”. These lenses can produce diffraction-limited spots by optimizing the width and arrangement of the slits with respect to the polarization direction. In many studies about SOLs, circularly or radially polarized waves are used for the incident beam and the slits, according to the direction of polarization, such that the design of the SOL consists of concentric circles and, thus, the shape of the focal beam spot is circular. If it is desired to focus the wave in only one direction, the lens can be designed with its slits aligned with the direction of polarization, and the principle is based on super-oscillation. We call this the cylindrical SOL.

In this paper, we describe a THz wave, cylindrical, super-oscillatory lens (THz-CSOL) numerically designed for industrial film inspection applications. In the inspection of products, such as films, defective sections with small metal pieces or other debris are discarded and are usually cut off perpendicular to the direction of the film travel. Detecting debris with high resolution along the direction of film travel offers the advantage of shortening the section to be discarded and reducing losses. Although there has been a previous study of components that focus light in only one direction, the focal beam width of 3 mm (3λ) was not below the diffraction limit and not quite sufficient for product inspection [38].

Figure 1 shows a schematic illustration of a product inspection process utilizing THz-CSOL. In the figure, a plastic film (e.g., polyethylene) is being moved along the *x*-direction by rollers. The THz-CSOL, which is characterized by slits in a metal layer, is installed at a focal length away from the product along the *z*-direction. The THz wave is polarized along the *x*-direction, and is focused by the THz-CSOL into a thin line with the focusing direction parallel to the short side of the slits or along the *x*-axis. When a dust particle (e.g., metal particle) is present on the product, the focused THz wave is subsequently diffracted. By collecting the reflected and diffracted wave using a one-dimensional detector array, the system can then detect and locate the dust particle. We numerically designed the THz-CSOL for 0.1 THz which has a corresponding longer wavelength than that in previous studies [38,41]. Based on numerical simulations, our THz-CSOL can achieve sub-diffraction resolution, long WD, and a line-shaped focus.

## 2. Methods

The numerical calculations for the THz-CSOL design and the THz wave intensity through the lens were performed in MATLAB. The THz-CSOL is designed to generate a thin line-shaped focal beam with a linearly polarized incident THz wave of 0.1 THz (wavelength λ = 3.0 mm). It has a focal length of 70 mm and has equal *x* and *y* dimensions of 100 mm (33.3 λ). A plane wave with the intensity profile of top hat type (*P*((−50 < *x* < 50, −50 < *y* < 50) = 1, *P*(x < −50, 50 < *x*, *y* < −50, 50 < *y*) = 0) was used as the incident beam. Here, *P*(*x*,*y*) is the intensity in xy coordinates. The beam size is the same size as the lens (100 mm × 100 mm). The direction of polarization is parallel to the *x* direction. These calculations were performed over a space of 400 mm × 400 mm. Perfectly absorbing spatial boundary conditions were used.

Figure 2a shows the top view of the THz-CSOL design, which is a symmetrical metal mask with slits. The black areas represent the parts with the metal layer (opaque), whereas the white regions represent the slits (transparent). The smallest slit width of our lens is 250 μm (*λ*/12), which is half the λ/6 width used in a previous study [41]. The smaller the width, the more freedom the design of SOL pattern has, which allows a narrower focal beam width. However, there is a lower limit for the slit width, because if it becomes too thin, the wave will not be able to pass through. Figure 2b depicts how the width and spacing of the slits, which are crucial parameters in the lens design, are derived. The entire lens surface is first divided into 200 unit regions, as shown in the figure. The transmittance of each unit region is then randomly assigned a value of 1 (transparent) or 0 (opaque) until the optimized transmittance pattern is achieved. The opaque condition is specified by setting the thickness of the metal layer above the skin depth to block the THz transmission completely. When fabricating SOLs, the pattern can be fabricated with a thin layer of metal, for example, gold, which has a skin depth of about 230 nm and can be deposited by sputtering process and photolithography or electron beam lithography [42]. Figure 2c shows the hexadecimal representation of the transmittance pattern from the center position toward the edge of the lens.

The THz-CSOL design was optimized using the full width at half maximum (FWHM), the intensity of the focal beam line, and the intensity of the side lobes as objective functions. Side lobes are considered because subwavelength localization with an SOL generates side lobes in the far field in addition to the central focal beam lime. At any point on the observation plane *xy* at a distance *z* after the lens, the intensity of the THz wave is calculated based on the following equations [43]: (1)$$E_{x}\left(x,z\right)=\int_{0}^{\infty }A\left(l\right)exp\left[j2\pi q\left(l\right)z\right]J_{0}\left(2\pi lx\right)2\pi ldl\;E_{z}\left(x,z\right)=-j\int_{0}^{\infty }\frac{l}{q\left(l\right)}A\left(l\right)exp\left[j2\pi q\left(l\right)z\right]J_{1}\left(2\pi lx\right)2\pi ldl$$
with
(2)$$A\left(l\right)=\int_{0}^{\infty }t\left(x\right)g\left(x\right)J_{0}\left(2\pi lx\right)2\pi xdx$$
where *A*(*l*) is the angular spectrum of the electric field in the lens pattern [44]; *j* is the imaginary unit; *q*(*l*) = (1/*λ*^2^ − *l*^2^)^1/2^ with *l* as the spatial frequency component in the *x* direction; *J*_0_ and *J*_1_ are the zeroth and first order Bessel functions, respectively; *t*(*x*) is the transmission function of the lens; and *g*(*x*) = *exp* (−*x*^2^/*w*_0_^2^), with *w*_0_ representing the waist radius of the Gaussian beam. When the SOL is illuminated by a linearly polarized beam, the total electric energy density can then be expressed as: *I*(*x*, *z*) = (|*E*_x_ (*x*, *z*)|^2^ + |*E*_z_ (*x*, *z*)|^2^). The optimization conditions require a focal beam width smaller than 0.9λ in the *x*-direction which is smaller than the diffraction limit of the conventional lens and a side lobe ratio (SLR) of less than 40%. SLR is the intensity ratio of the maximum side lobe to the central lobe of the beam line. Large side lobes, which prevent achieving fine measurement resolutions, are undesirable. Hence, optimization of the lens design is necessary for achieving both a narrow central peak width and a small SLR.

In addition to the desired focal length of *z* = 70 mm, the THz-CSOL was also optimized to satisfy the conditions at *z* = 68, 69, 71, and 72 mm. The optimization algorithm used was binary particle swarm optimization (BPSO) [45]. The algorithm flowchart is shown in Figure 3.

## 3. Results and Discussion

The beam characteristics and intensity profiles of the THz wave transmitted by the numerically designed and optimized CSOL are presented in Figure 4, where the *z*-axis is the propagation direction and *xy* plane is the focal plane. Figure 4a,b show the intensity distribution after the THz wave passes through the lens, normalized by the maximum intensity of main peak at *z* = 71.2 mm. Figure 4a shows the normalized intensity pattern of the focused THz beam on the *xz* plane and a hotspot is observed from *z* = 67.7 mm to *z* = 76.6 mm with a maximum intensity at 71.2 mm. On the other hand, Figure 4b shows the intensity pattern on the *xy* plane. The focal spot of the transmitted THz beam exhibits a constant maximum intensity along the *y* direction, which is perpendicular to the polarization. We considered an incident beam that is 100% polarized in one direction. Terahertz waves polarized along the direction parallel to the length of the slits are reflected, while terahertz waves that are polarized along the widths of the slits are transmitted. Side lobes are seen at around *x* = ±6.1 mm from the central focal beam line at *x* = 0. The intensity profile at *y* = 0 mm and *z* = 70 mm, which is the target focal length for the designed CSOL, is shown as a function of *x* in Figure 4c. It can be observed in this figure that the beam width is 0.84λ and the SLR is about 30%. We note that the ratio of the THz power transmitted through the CSOL to the total power of the incident THz beam is 25.5%. This implies a relatively low transmittance, especially in comparison to conventional lenses. Improving the power of the focused beam line is one of the future challenges of designing the CSOL.

Figure 5a shows the spatial distribution of the width of the focal beam line and the SLR along the *z* direction. The red plot shows the beam widths normalized by the wavelength (3 mm) of the incident wave while the cyan plot represents the SLR values. Figure 5a also shows the diffraction limit *δ* (solid black line), which is calculated as
(3)δ=0.61λ/NA
where the NA is the numerical aperture of the CSOL. The NA is calculated as
NA = *n* sin*θ*(4)
where *n* is the refractive index of air (the medium surrounding the lens) while *θ* is the maximum half-angle of the light cone that is defined by an apex at the focal point and a base on the lens.

As can be seen in Figure 5a, the minimum width of the focal beam line is 0.83λ (2.5 mm) at *z* = 71.2 mm, where the SLR is also around the minimum at 32.9% which satisfies our BPSO optimization criterion of SLR below 40%. We note that the 0.83λ beam width is ~21% smaller than the diffraction limit. Moreover, from *z* = 67.7 mm to 76.6 mm, the resolution of the lens is better than the diffraction limit. In addition, the effective NA of the 100 mm × 100 mm CSOL with a 70-mm focal length is 1.43, which is noteworthy because NA over 1 cannot be achieved with conventional lenses. This result also indicates that the DOF is 10 mm (3.3 λ), which is a little longer than the 8.8-mm DOF of conventional lenses. Here, the DOF is calculated as λ/NA^2^, in the same way as that of conventional lenses [46]. In the case of conventional lenses, the longer the DOF, the smaller the NA, but this does not apply to the THz-CSOL. Thus, the CSOL is unique for having comparable DOF but greater than 1 NA.

Figure 5b shows the intensity profile along the *z* direction. With maximum at *z* = 71.2 mm, the intensity is at least 68.4% from z = 67.7 mm to 76.6 mm, when the focal beam width is smaller than the diffraction limit.

The influence of the size of the incident beam on the profile and intensity of the transmitted THz beam through the CSOL were also investigated. Figure 6a–c show the incident beam profiles calculated using the 1st order Bessel function *J*_0_(*s*), where *s* = *x*/2 was used for the plots in Figure 6a,d, *s* = *x*/2.5 was used for those in Figure 6b,e, and *s* = *x*/3 was used for those in Figure 6c,f. The intensity patterns at the focal length (*z* = 70 mm) are presented in Figure 6d–f, which collectively show that the broader the incident beam profile, the smaller the beam width and the SLR. These results also indicate that the outer parts of the lens strongly influence the one-dimensional focusing, which results in thin beam line generation. Typical THz sources, such as the IMPATT diode, the Gunn diode, and the resonant tunneling diode, do not transmit only at a specific frequency, but rather at multiple frequencies [47,48,49]. It is then reasonable to expect that the THz-CSOL performance will vary depending on the THz wave frequencies. Hence, we confirmed the frequency dependence of the lens. Figure 7a shows a comparison of the focused beam widths when the THz-CSOL, which is optimized for 0.1 THz, is used at slightly different frequencies. The results show that the focal length of the lens is shortened by 1 mm when the frequency is 0.098 THz. On the other hand, its focal length becomes 2 mm longer when the frequency is 1.02 THz. In other words, the focal length of the THz-CSOL decreases with lower frequency and increases with higher frequency. This is the opposite of what is expected with glass- or plastic-type conventional lenses, which have shorter focal lengths at higher frequencies due to the frequency dependence of their refractive indices. In the case of the THz-CSOL, as the frequency of incident beam becomes smaller, the wavelength becomes longer relative to the widths of the slits and thus the diffraction effects are stronger. This could be one of the reasons why the focal length is shortened. In Figure 7b, the focal beam widths at *z* = 70 mm and the corresponding SLR values are shown when the CSOL is used at different frequencies. The farther away from the frequency used for optimization, the wider the width and the weaker the focusing. This graph confirms that both the focal beam width and the SLR are optimized to be small at 0.1 THz. This graph also shows that the SLR stays below 35% in the range between 0.093 THz to 0.103 THz. Moreover, the focal beam width remains around 0.9λ between 0.093 THz to 0.102 THz. Thus, these results show that the frequency range for the ideal performance of the lens is tightly limited within less than 0.01 THz from the optimal 0.1 THz, which is comparable to the frequency bandwidth of commonly used CW THz sources, such as the IMPATT diode. In other words, the THz-CSOL is not optimized for a wide frequency range. However, it is possible to design a SOL that can generate a focal beam for a wide frequency range by optimizing it at multiple frequencies.

## 4. Conclusions

In this study, we designed by numerical approach a THz-CSOL optimized for 0.1 THz for industrial applications. We checked, in principle, whether it is possible to design a SOL that focuses the THz wave in one direction. Simulated results show that a linearly polarized THz wave incident on the THz-CSOL is transmitted through the optimized device as a line-shaped THz radiation with a focal beam width that is thinner than the diffraction limit on the far-field. With the thin beam output, the THz-CSOL can be conveniently used as a key component in a product inspection process with short inspection time. The optimized design of the CSOL has a size of 100 mm × 100 mm and generates a thin focal beam line width of 0.84λ, which is 21% smaller than the diffraction limit, at a focal length of 70 mm (23.3λ). It has an effective NA of 1.43 and DOF of 3.3λ. By comparing the results for different incident beam profile sizes, we also showed that the outer part of the lens is important for generating a thin focal beam line. In this study, we optimized the lens using plane THz waves although THz sources often have Gaussian output beams. However, it is also possible to design the CSOL for typical THz beams by optimizing it using THz waves with Gaussian intensity distribution as the incident radiation. Our proposed THz-CSOL, which meets the requirements for industrial inspections, is practical because its dimensions can be easily modified to suit the size of the film product that is going to be tested. It is anticipated to become one of the powerful tools for product inspection using THz waves.

## Figures and Tables

**Figure 1 sensors-21-06732-f001:**
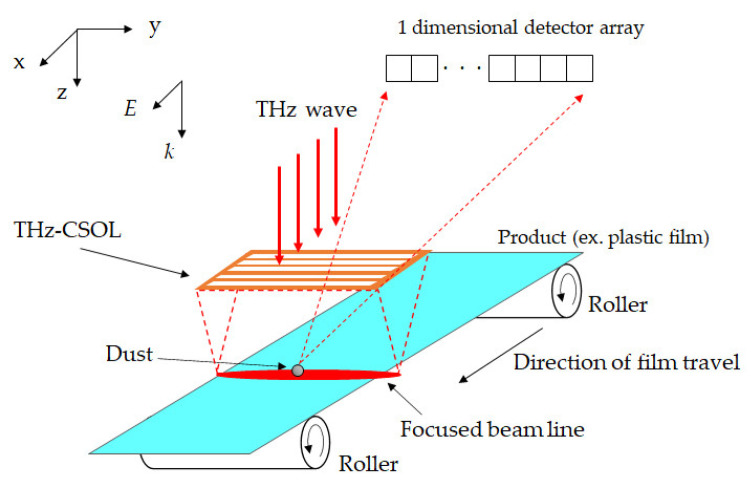
Schematic illustration of product inspection with THz cylindrical super-oscillatory lens (THz-CSOL). The propagation direction *k* and polarization direction *E* of the THz wave are parallel to the *z* and *x* axes, respectively. (Not drawn to scale.).

**Figure 2 sensors-21-06732-f002:**
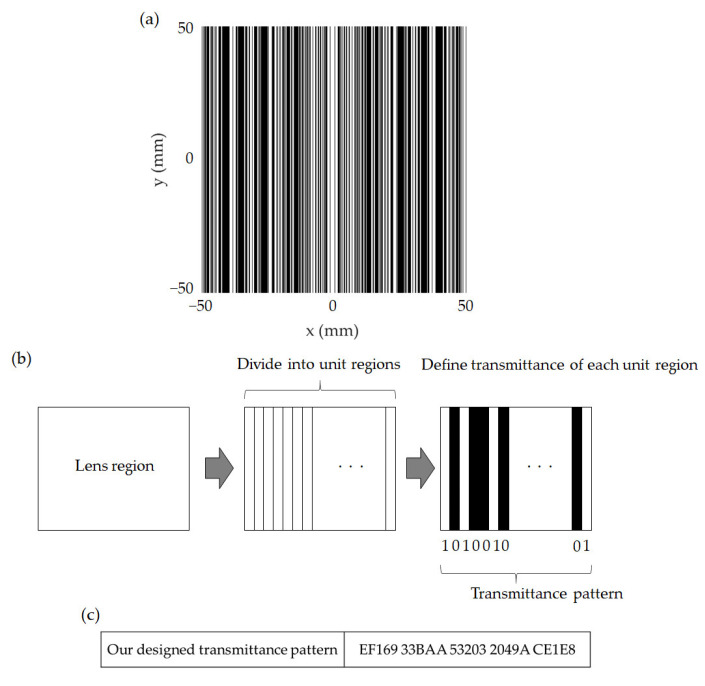
(**a**) Schematic of our designed THz-CSOL where the black and white regions correspond to opaque and transparent areas, respectively. (**b**) Steps of the transmittance pattern design process. (**c**) Hexadecimal representation of the transmittance pattern of our symmetrically designed lens from the center position to the edge of the lens.

**Figure 3 sensors-21-06732-f003:**
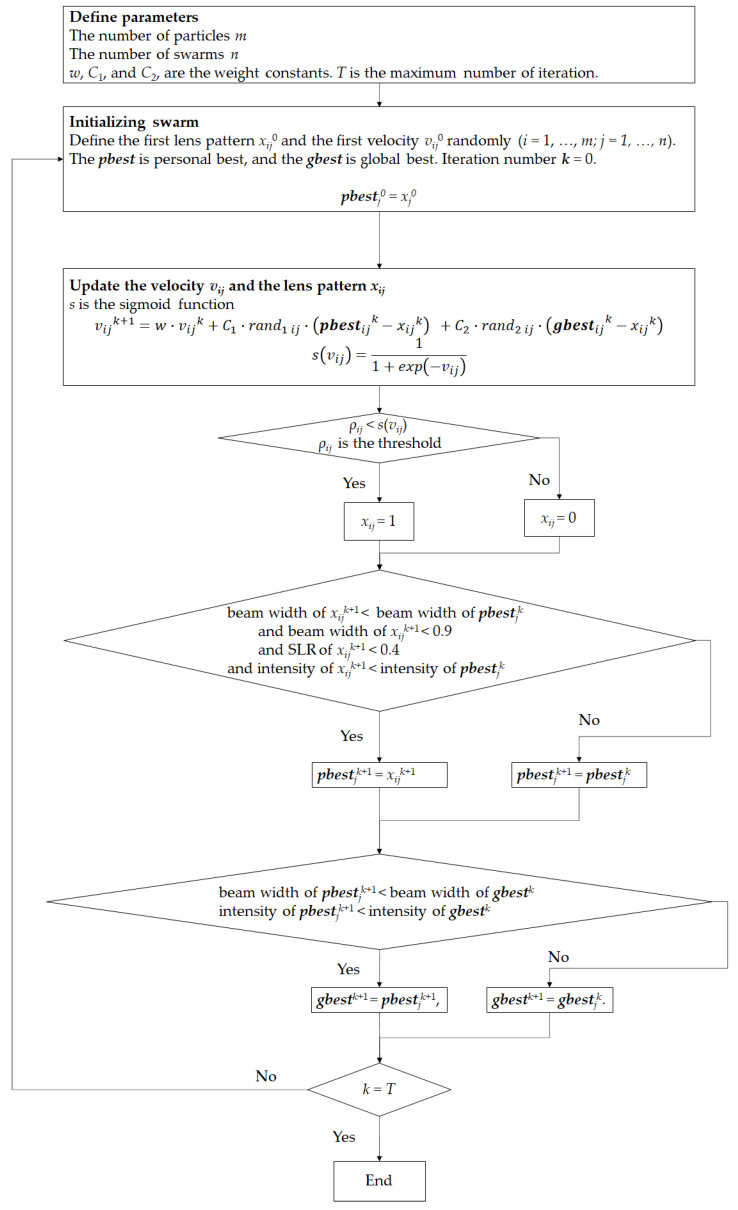
Algorithm flowchart of optimizing the lens pattern using binary particle swarm optimization (BPSO). The parameters are defined as *m* = 200, *n* = 100, *w* = 0.5, *C_1_* = *C_2_* = 1, and *T* = 1000.

**Figure 4 sensors-21-06732-f004:**
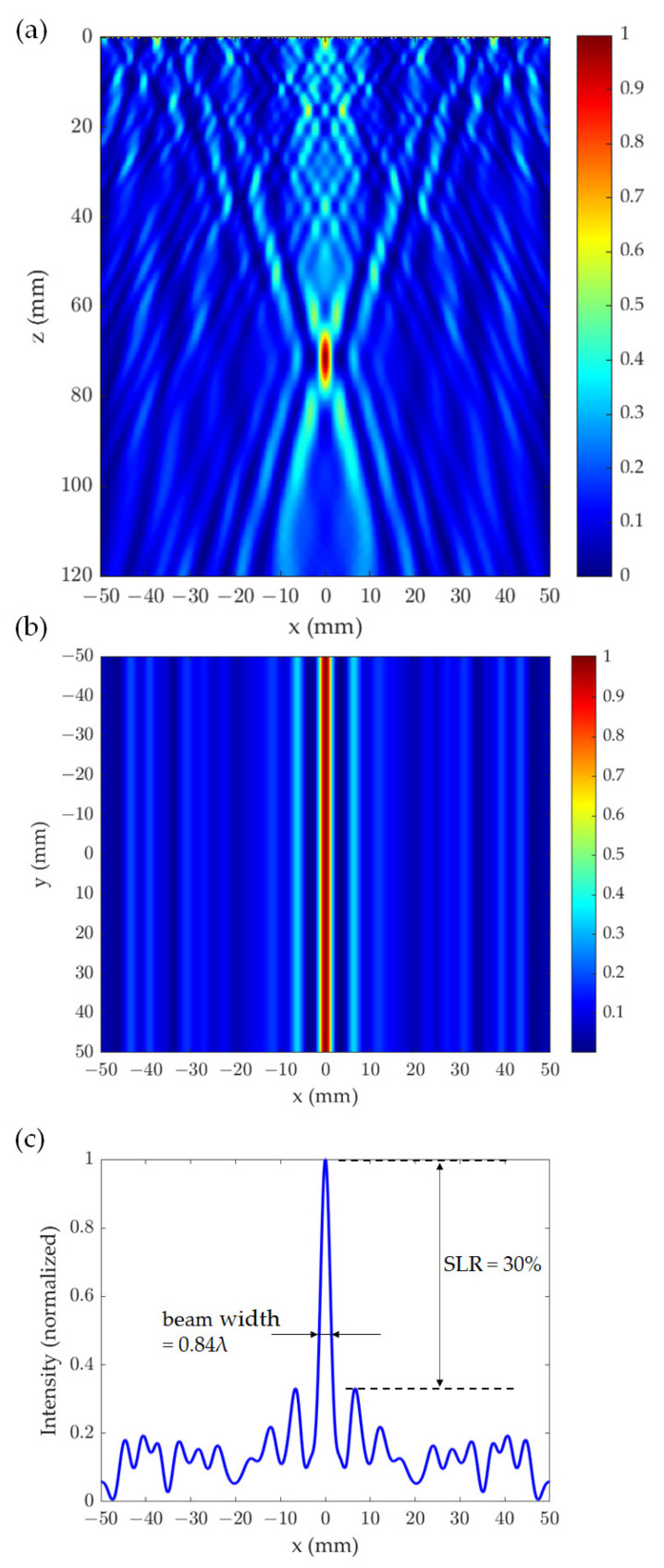
Characteristics of the transmitted THz beam by the cylindrical super oscillatory lens (CSOL), which was optimized for 0.1 THz and desgined with a focal length of 70 mm. (**a**) Normalized intensity pattern on the *xz* plane at *y* = 0 mm; (**b**) normalized intensity pattern at *z* = 70 mm on the *xy* plane, which is the focal plane; and (**c**) the intensity profile along the *x* direction at *y* = 0 mm and *z* = 70 mm.

**Figure 5 sensors-21-06732-f005:**
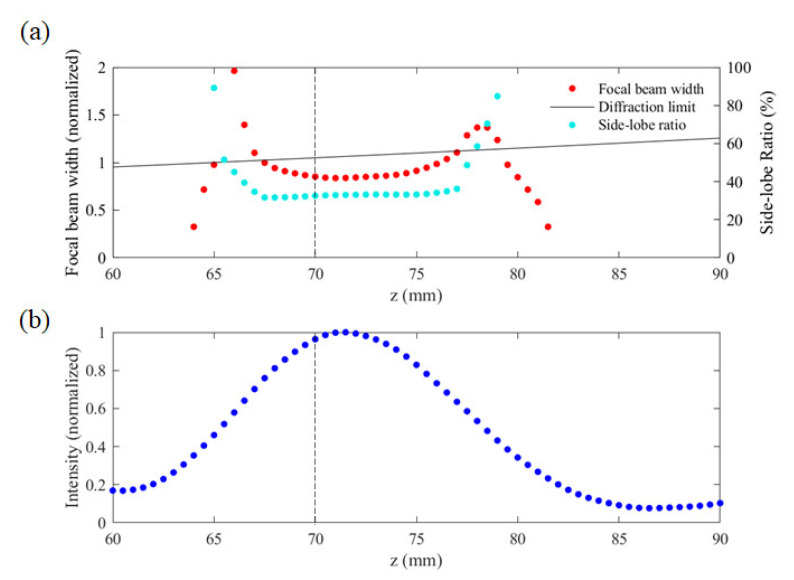
(**a**) The focal beam width along the z direction. The beam width values (red points) are normalized to the incident wavelength of 3 mm. The black solid line represents the diffraction limit while cyan points represent the side-lobe ratio; (**b**) The intensity profile of the focused THz beam at *x* = 0 mm, where the intensity values are normalized to the maximun intensity. The dashed line in (**a**,**b**) indicates the designed focal length of 70 mm.

**Figure 6 sensors-21-06732-f006:**
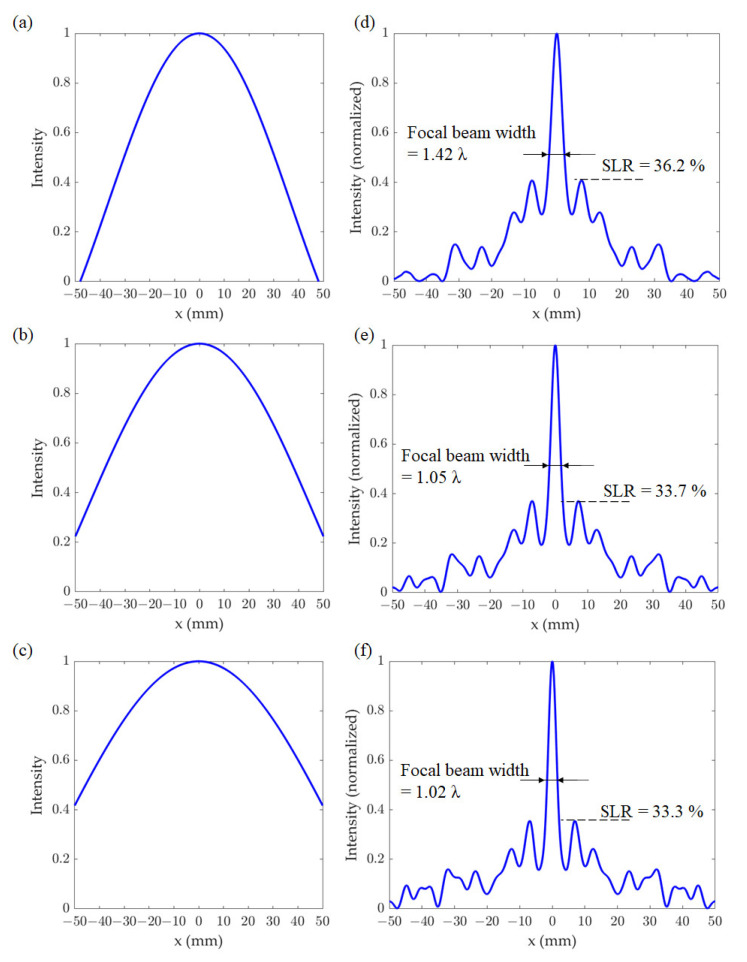
Incident beam profiles and the corresponding focal beam lines generated by the THz-CSOL. (**a**–**c**) IThe simulated incident beam profiles; and (**d**–**f**) the calculated intensity profiles along the x direction at the focal length (*z* = 70 mm).

**Figure 7 sensors-21-06732-f007:**
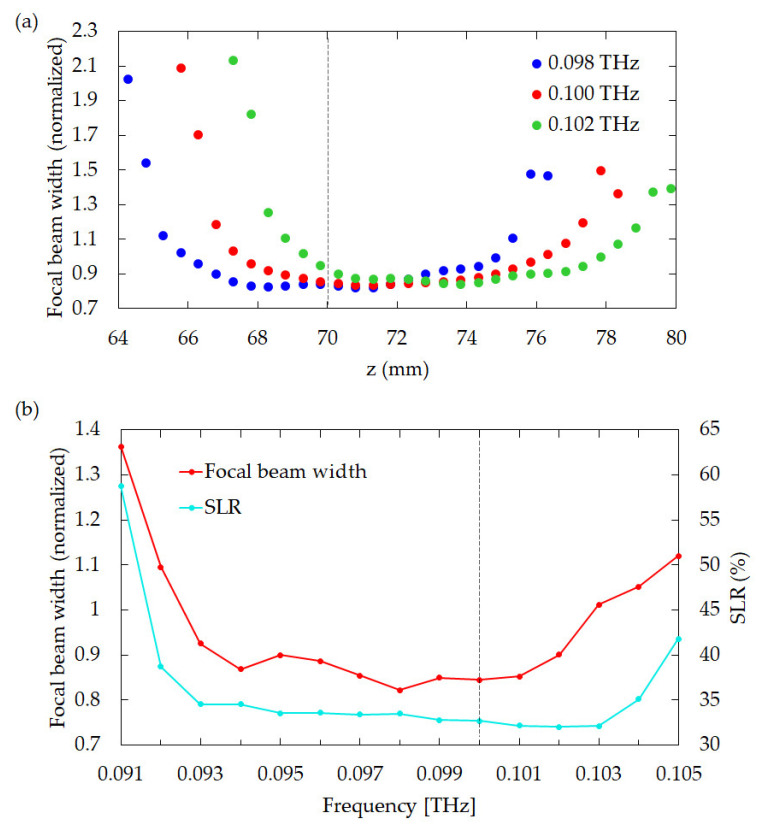
Frequency dependence of the THz-CSOL. (**a**) Beam widths of the transmitted THz wave by the CSOL at different frequencies: 0.098 THz (blue), 0.100 THz (red), and 0.102 THz (green). The black dashed line is the designed focal length, 70 mm. (**b**) Comparison of beam widths at *z* = 70 mm when the CSOL is used for a range of different frequencies (0.095 THz – 0.105 THz). Red dots with lines represent the normalized beam widths while cyan-colored dots with lines represent the SLR. The black dashed line indicates the frequency (0.100 THz) for which the design of the CSOL was optimized.

## Data Availability

The data presented in this study are available on request from the corresponding author.

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
