# Peer review of "A Study of Terahertz-Wave Cylindrical Super-Oscillatory Lens for Industrial Applications"

_sensors, 2021, doi:10.3390/s21206732_

Round 1
Reviewer 1 Report
The paper describes an interesting approach to optics for inspecting objects on conveyor belts. The described CSOL lead to a focus which is in a direction below the diffraction limit and has a larger DOF than classical lenses. From this point of view, this is very interesting.
However, the authors write nothing about the intensity distribution along the line focus. Only a normalized line is shown here (Fig. 4b.) Normalized to what? The authors should also comment on what is the benefit of optics for the specified application that greatly improves focus in one direction but does not focus in the other direction, making the pixel arrangement of the detector array critical.
In the discussion of Fig.7, it is only explained that the focus is frequency dependent for a fixed z-position. It should also be explained why the focus position (=location of the smallest waist) moves with frequency.
Reviewer 2 Report
Dear Authors,
In my opinion, the presented manuscript could be interesting for the community and readers of Sensors. However, my overall recommendation is “Reconsider after major revision (control missing in some experiments)”.
- I am not sure, if the name cylindrical super-oscillatory lens is correctly used in this manuscript. Usually, in the literature CSOLs consists of many concentric rings. In your case, linear shape of the metallic stripes is considered.
- It is suggested to add and maybe compared other similar lens solutions used for the THz scanners to obtain the linear focus, e.g. “3-D-printed flat optics for THz linear scanners” J Suszek et al, IEEE transactions on Terahertz Science and Technology 5 (2), 314-316.
- It is unclear, what is the distribution of the THz beam incident on the CSOL. Is it a plane wave? Normally THz sources have Gaussian (or similar) output beam. Have you considered the problem how to transform the circular beam from a source to fit the requirement of shining the rectangular CSOL?
- I think that such a metallic lens has attenuation due to unwanted reflection and diffraction. Can you comment on power efficiency of the CSOL, I mean the ratio - total power incident on the film in the main peak to total power incident on the CSOL.
- It is unclear what kind of film you plan to inspect using the CSOL. Is 10 cm width enough to inspect such films? Can you increase the length of the beam in focus?
- You probably considered only beam with 100% polarization along y axis. What about the real situation when not all radiation is perfectly linear? How is the lens sensitive to the incident beam polarization adjustment?
- “The smallest slit width of our lens is 250 μm (λ/12)”. Why did you use such value? What happens if you use e.g. 125 um. Does it improve the focus FWHM?
- “The opaque condition is specified by setting the thickness of the metal layer above the skin depth to block the THz transmission completely”. What was the thickness in the simulation, e.g. 100 um? Is the CLSO with such a thickness easy to manufacture?
- It is recommended to add in the conclusion a paragraph analysing feasibility of CSOL manufacturing. Can CSOL be manufactured using a simple process or required a complicated technology?
- Can you comment in conclusions not only on advantages (e.g. thin focal beam width) but also on disadvantages of your lens, e.g. can work only with polarized sources.
Round 2
Reviewer 2 Report
Dear Authors,
I think that You answered all my remarks/comments. Thank you. In my opinion the paper is ready to publish. My recommendation is “Accept in present form”.